# Transcription reshapes RNA hairpin folding pathways revealed by all-atom molecular dynamics simulations

Peng Tao[1,2*☯], Yunda Si[1☯], Jie Xia[3,4☯], Yi Xiao[2]

**1** Key Laboratory of Systems Health Science of Zhejiang Province, School of Life Science, Hangzhou Institute for Advanced Study, Hangzhou; University of Chinese Academy of Sciences, Beijing, China, **2** School of Physics, Huazhong University of Science and Technology, Wuhan, China, **3** College of Information Engineering, Zhejiang University of Technology, HangZhou, China, **4** State Key Laboratory of Cell Biology, Shanghai Institute of Biochemistry and Cell Biology, Center for Excellence in Molecular Cell Science, Chinese Academy of Sciences, Shanghai, China

☯ These authors contributed equally to this work.
* taopeng@ucas.ac.cn

## Abstract

The divergence in folding pathways between RNA co-transcriptional folding (CTF) and free folding (FF) is crucial for understanding dynamic functional regulation of RNAs. Here, we developed a simplified all-atom molecular dynamics framework to systematically compare the folding kinetics of an RNA hairpin (PDB:1ZIH) under CTF and FF conditions. By analyzing over 800 microseconds of simulated trajectory, we found that despite convergence to identical native conformations across CTF simulations (with varied transcription rates) and FF simulations, they exhibit distinct preferences for the folding pathways defined by the order of base-pair formation. Conformational space analysis shows that CTF biases the folding pathway by adopting more compact conformations than FF. Our findings provide atomic-scale insights into how temporal-spatial coupling of transcription and folding diversifies RNA folding dynamics.

## Author summary

In living cells, RNA molecules begin folding while they are still being synthesized—a process known as co-transcriptional folding. This differs fundamentally from how RNA folds in test tubes, where synthesis is complete before folding starts. Using advanced computer simulations at atomic detail, we discovered that how an RNA hairpin folds depends critically on whether it folds during or after synthesis. Specifically, we simulated a small RNA hairpin under both scenarios. While it always reached the same final folded shape, the folding pathway—the sequence of steps it took to get there—diverged dramatically. During co-transcriptional folding, the RNA preferred pathways where distant base pairs

**Data availability statement:** The code and data of CTF simulations are available at https://github.com/PengTao-HUST/RNA_CTF.

**Funding:** This work is supported by the National Natural Science Foundation of China (11874162 to YX). The funders had no role in study design, data collection and analysis, decision to publish, or preparation of the manuscript.

**Competing interests:** The authors have declared that no competing interests exist.

formed first, whereas free folding favored starting near the loop region. Our work reveals that the spatiotemporal coupling of transcription and folding reshapes RNA energy landscapes, biasing the molecule toward otherwise rare folding routes. This provides a mechanistic explanation for how cellular environments diversify RNA behavior and underscores why studying RNA folding in vivo is essential to understand its biological functions. By bridging atomic-scale dynamics with cellular contexts, we offer new principles for RNA-targeted therapeutic design.

## 1 Introduction

The AlphaFold [1] series, as a representative of deep learning methods, has preliminarily addressed the challenge of protein structure prediction. However, current approaches remain far from successful in resolving RNA structure prediction, with one fundamental reason being the more complex folding dynamics of RNA compared to proteins. Historically constrained by experimental and computational limitations, research on RNA folding mechanisms has predominantly focused on *in vitro* systems where synthesized RNA molecules fold in optimized solution environments [2]. Nevertheless, it should be emphasized that *in vivo* RNA folding occurs co-transcriptionally on RNA polymerase during chain elongation, a process termed co-transcriptional folding [3] (CTF). As this process is kinetically controlled, its folding products and pathways may substantially differ from those of thermodynamically controlled refolding processes. Furthermore, multiple physiological parameters including transcription elongation rates [4], site-specific transcriptional pauses [5], and ionic variations [6] could collectively influence RNA folding behavior in cellular contexts. Therefore, investigating the mechanisms of CTF is crucial for understanding RNA functions [7], particularly given that the folding dynamics during transcription may directly determine RNA's biological activity.

Recent years have witnessed significant advancements in experimental and computational methodologies for investigating CTF. Experimentally, Frieda et al. [8] pioneered the direct observation of CTF in the adenine riboswitch using single-molecule optical tweezers. Uhm et al. [9] employed single-molecule Förster resonance energy transfer (FRET) to study the thiamine pyrophosphate riboswitch, revealing that transcriptional pausing near the translation initiation codon regulates its conformational switching. Lou et al. [10] utilized single-molecule FRET to demonstrate that the glmS ribozyme undergoes self-cleavage during transcription prior to full-length synthesis. Watters et al. [11] introduced SHAPE-seq, achieving nucleotide-resolution monitoring of CTF dynamics. Incarnato et al. [12] identified coexisting cooperative and non-cooperative folding events during RNA transcription via SPET-seq. Recently, Yu et al. [13] leveraged eSPET-seq to elucidate widespread structural transitions and the anti-mutator effect of nascent RNA folding during transcription. Szyjka et al. [14] developed TECprobe-VL, a high-resolution co-transcriptional RNA chemical probing method, which identified coordinated folding events mediating transcription antitermination.

Computationally, Lutz et al. [15] combined coarse-grained molecular dynamics (MD) and Monte Carlo (MC) simulations to analyze CTF pathways of SAM-I and adenine riboswitches, demonstrating divergence from their refolding trajectories under equilibrium conditions. Wang et al. [16] employed conformational resampling with kinetic relaxation to study CTF in E. coli SRP RNA and HIV-1 TAR RNA, revealing that transcriptional pauses drive efficient conformational rearrangements to protect functional architectures and modulate folding pathways. To enhance sampling efficiency, Xu et al. [17] transformed RNA folding energy landscapes into discrete partition-network representations, subsequently simulating length-dependent folding behaviors via kinetic MC simulations and master equation to predict CTF patterns. Recently, Jin et al. [18] integrated computational simulations with NMR spectroscopy data, identifying different folding pathways responsible for transcriptional pausing and elucidating folding mechanisms involving non-native intermediates that trigger transcriptional pause.

Despite significant advancements in existing research [19–21], the atomic-scale distinctions between RNA CTF *in vivo* and refolding or free folding (FF) *in vitro* remain unresolved. Notably, critical questions persist: whether folding pathways are identical between the CTF and FF, if divergent, the underlying physical mechanisms driving such differences remain elusive. While all-atom MD simulations have been extensively employed to investigate RNA structural dynamics [22–24], simulating CTF at atomic resolution faces formidable challenges. These arise from the extended timescale of CTF (spanning milliseconds to minutes) and the intricate interplay between nascent RNA chain and RNA polymerase [25].

In this study, we focus specifically on how the co-transcriptional nature of RNA synthesis, i.e., folding occurs concurrently with transcription, impacts RNA folding kinetics. To address this, we developed a simplified all-atom MD framework for simulating CTF by introducing a movable rigid plane to prevent non-physical interactions between synthesized and unsynthesized nucleotides (Fig 1b). Using this framework, we investigated the CTF mechanism of an RNA hairpin (PDB ID: 1ZIH, Fig 1a) and compared it with conventional FF simulations. We found that while 1ZIH exhibits identical folding conformations across CTF simulations with varying transcription elongation rates and FF simulations, CTF and FF exhibit distinct preferences for the folding pathways. These results demonstrate that the spatiotemporal coupling of transcription and folding can reshape RNA folding pathways, providing mechanistic insights into how RNA transcription may diversify folding dynamics.

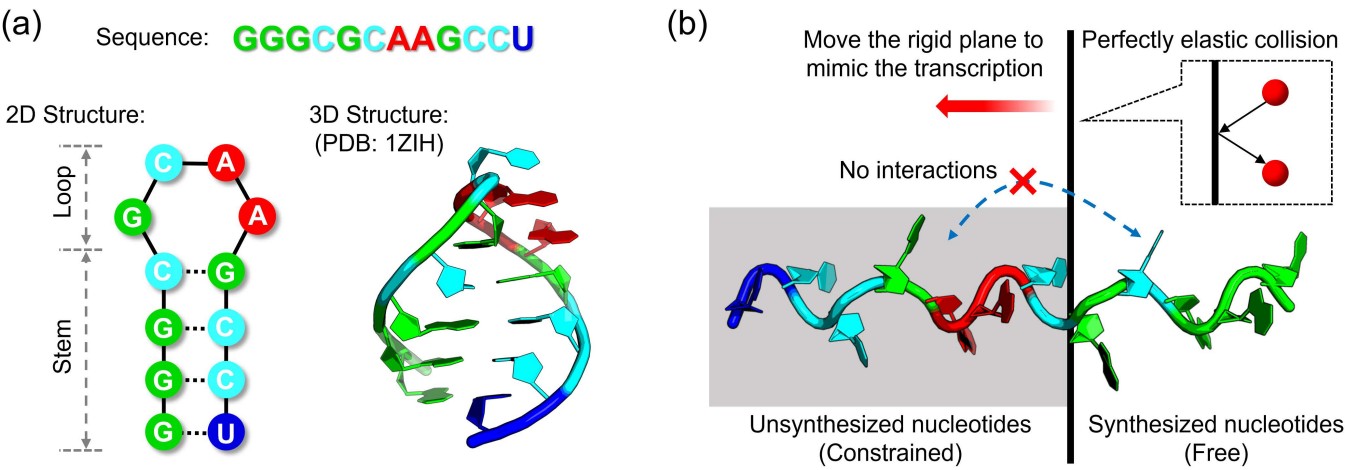

**Fig 1. Simulated system and methodology.** (a) Primary, secondary, and tertiary structures of the RNA hairpin molecule 1ZIH. The native structure comprises a stem region containing three C-G pairs and one G-U pair, connected to a loop region. (b) Simplified co-transcriptional folding (CTF) simulation framework. A movable rigid plane is introduced to separate synthesized nucleotides (right side; free to move) from unsynthesized nucleotides (left side; constrained by a harmonic oscillator potential), with perfectly elastic collisions occurring between right-side atoms and the plane. This design mimics the progressive nucleotide addition during transcription.

## 2 Methods

### 2.1 Simulation system and force field

The RNA hairpin 1ZIH (Fig 1a) contains 12 nucleotides (nt; sequence: 5'-GGGCGCAAGCCU-3') stabilized by three Watson-Crick pairs (4C-9G, 10C-3G, 11C-2G) and a wobble 1G-12U pair. The initial extended single-strand conformation (as shown in Fig 1b) of 1ZIH was built by the *tleap* program in the Amber18 package [26]. All simulations employed the AMBER ff99bsc0+χOL3 RNA force field [27], consisting of a sugar-phosphate parameter refinement and a pucker flipping modification. Systems were solvated in TIP3P water with a buffer size of 12 Å (this is appropriate for this study, see the comparison of the buffer size of 15 Å in S1 Fig), neutralized with K+ ions, and supplemented with 0.3 M KCl. Periodic boundary conditions were applied in all simulations. The water box dimensions are [81.1 Å × 47.1 Å × 56.2 Å]. Ion parameters adopted the TIP3P-optimized set from Li et al. [28], featuring 12–6 Lennard-Jones potentials calibrated to experimental hydration free energies, demonstrating enhanced transferability in ionic solutions.

### 2.2 Minimization, heating and equilibration

Prior to MD production, each trajectory was prepared through two minimization steps and one heating step. Specifically, the first step involved minimizing the buffer water and ions for 4000 steps, with the atoms in the RNA hairpin being fixed. The second step entailed minimizing the entire system without any restraints for 10000 steps. Finally, the system was heated from 0 to 300 K using an NVT ensemble, with the seed for the pseudorandom number generator determined by the running date and time of the simulation. Following these preparatory steps, a 200-ps preproduction stage was equilibrated. In the production stage, the SHAKE [29] algorithm was employed to bind hydrogen atoms to their neighboring heavy atoms, thereby enabling a 2-fs time step. The temperature was maintained at 300 K by coupling to a Langevin heat bath with a frequency of 2 ps, while the pressure was kept around 1 bar using a Berendsen barostat [30] with isotropic position scaling. Nonbonded interactions were truncated at 12 Å, and long-range electrostatic interactions were calculated using the particle mesh Ewald [31] with a grid spacing of 1 Å. The trajectory was recorded every 10000 steps, corresponding to 20 ps per frame.

### 2.3 FF and CTF simulations

In the CTF simulations, the extended 1ZIH chain was oriented along the x-axis, with the 5' end pointing in the positive direction. To restrain the motion of all residues, a harmonic oscillator potential with a strength of 10 kcal/mol Å² was applied. To simulate the process of folding during transcription, a rigid plane perpendicular to the x-axis was introduced to prevent interactions between synthesized and unsynthesized nucleotides. When RNA atoms on the right side of the plane collided with it, a perfectly elastic collision occurred, meaning that the tangential component of the velocity remained unchanged while the normal component was reversed. At the beginning of the CTF simulations, the plane was placed at the left of the first nucleotide, and the constraint on this nucleotide was removed to allow it to move freely (equivalent to the completion of its synthesis). At intervals of Δt, the plane was moved one nucleotide to the left, and the constraint on that nucleotide was simultaneously removed. After 11Δt, when the plane reached the last nucleotide and all constraints were removed, the plane was also removed, transitioning the CTF into FF simulations. To account for the effect of transcription speed on the simulation results, CTF was simulated at three different speeds, with Δt set to 1, 10, and 100 ns, respectively denoted as CTF-1, CTF-10, and CTF-100. For each transcription speed, 12 independent trajectories were simulated, each approximately 15 μs in length. In the FF simulations, the same initial structure used in the CTF simulations was employed, and a total of 60 independent trajectories were simulated, each lasting 4–5 μs (Table 1). The program pmemd.cuda [32] was used to generate all the trajectories on the Nvidia RTX 3080Ti GPU. The code for the CTF simulations was written in Fortran and integrated into the source code of pmemd.cuda.

PLOS Computational Biology

**Table 1. Summary of the different types of simulations in this study.**

| Simulation type | Number of independent trajectories | Simulation time (µs) | Total simulation time (µs) |
|---|---|---|---|
| FF | 60 | 4-5 | 271.9 |
| CTF-1 | 12 | ~15 | 180.8 |
| CTF-10 | 12 | ~15 | 181.0 |
| CTF-100 | 12 | ~15 | 181.2 |
| Total | | | 814.9 |

## 2.4 Trajectory analysis

To define the folding events of 1ZIH, we employed two analytical metrics: the root-mean-square deviation (RMSD) and the number of hydrogen bonds. The RMSD serves as a measure of the similarity between the simulated structure and the native structure, with its calculation formula given by

$$RMSD = \sqrt{\frac{1}{N}\sum_{i=1}^{N}\left(x_i - x_i^{nat}\right)^2 + \left(y_i - y_i^{nat}\right)^2 + \left(z_i - z_i^{nat}\right)^2}$$

where $N$ is the number of atoms, $(x_i, y_i, z_i)$ and $(x_i^{nat}, y_i^{nat}, z_i^{nat})$ are the coordination of the $i$-th atom in simulated and native conformation, respectively. Owing to the high flexibility of the loop region in 1ZIH, only the heavy atoms in the stem region were considered when calculating the RMSD. A value of RMSD less than 2 Å was taken as an indication that 1ZIH had folded into its native state. Furthermore, the native structure of 1ZIH comprises 4 base pairs, with three C-G pairs contributing 9 hydrogen bonds and one G-U pair contributing 2 hydrogen bonds. The formation of these 4 base pairs or 11 hydrogen bonds was also regarded as indicative of 1ZIH attaining its native conformation. The criteria for hydrogen bond formation were based on a maximum donor-acceptor distance of 3.5 Å and a minimum donor-proton-acceptor angle of 120°, corresponding to the default geometric parameters in cpptraj [33], the trajectory analysis tool in Amber. Notably, we did not adopt a strict definition of folding in this work (i.e., simultaneous formation of the stem and loop), because only 14 trajectories successfully folded the stem. If we further required successful folding of the loop, the number of strictly successful folding trajectories would be reduced to 5 (a representative example is shown in S2 Fig), making subsequent analysis impossible. Lastly, in the construction of the conformational space, we computed the radius of gyration (RoG) of 1ZIH, with its calculation formula given by

$$RoG = \sqrt{\frac{1}{N}\sum_{i=1}^{N}\left(x_i - \overline{x_i}\right)^2 + \left(y_i - \overline{y_i}\right)^2 + \left(z_i - \overline{z_i}\right)^2}$$

where $(\overline{x_i}, \overline{y_i}, \overline{z_i})$ is the mass center of the conformation. All analyses were performed by cpptraj [33]. Structural snapshots were visualized by pymol [34].

## 3 Results

### 3.1 Native-state folding of 1ZIH in free folding simulations

As a MD-based study, validation of force field reliability is prerequisite. A critical benchmark is whether an RNA molecule can attain its native conformation within sufficient simulation duration. For 1ZIH, an RNA hairpin comprising a stem region and a flexible loop, we focused only on stem-region heavy-atom root-mean-square deviation (RMSD) relative to the native structure as the folding criterion, considering RMSD values below 2 Å indicative of successful folding events.

Starting from an extended initial conformation, we performed 60 independent folding trajectories, among which 5 achieved native-state folding (S3 Fig). Fig 2a illustrates a representative folding trajectory, demonstrating stable mainte-nance of stem-region RMSD within 2 Å after folding (structural alignment shown in Fig 2b), confirming native-state stability under the applied force field and simulation parameters.

Complementary to RMSD analysis, we evaluated folding events through formation of four base pairs: three GC pairs and one GU pair, containing three and two hydrogen bonds respectively (Fig 2c). Native-state attainment was defined by simultaneous formation of all 11 hydrogen bonds. Fig 2d maps the formation of these hydrogen bonds along the trajectory depicted in Fig 2a (see S4 Fig for the others).

## 3.2 Comparative analysis of co-transcriptional folding across elongation rates versus free folding

To systematically compare CTF and FF mechanisms of 1ZIH, we simulated CTF processes at three distinct elongation rates (1, 10, and 100 ns/nt, denoted as CTF-1, CTF-10, and CTF-100), with 12 independent trajectories per rate. Across

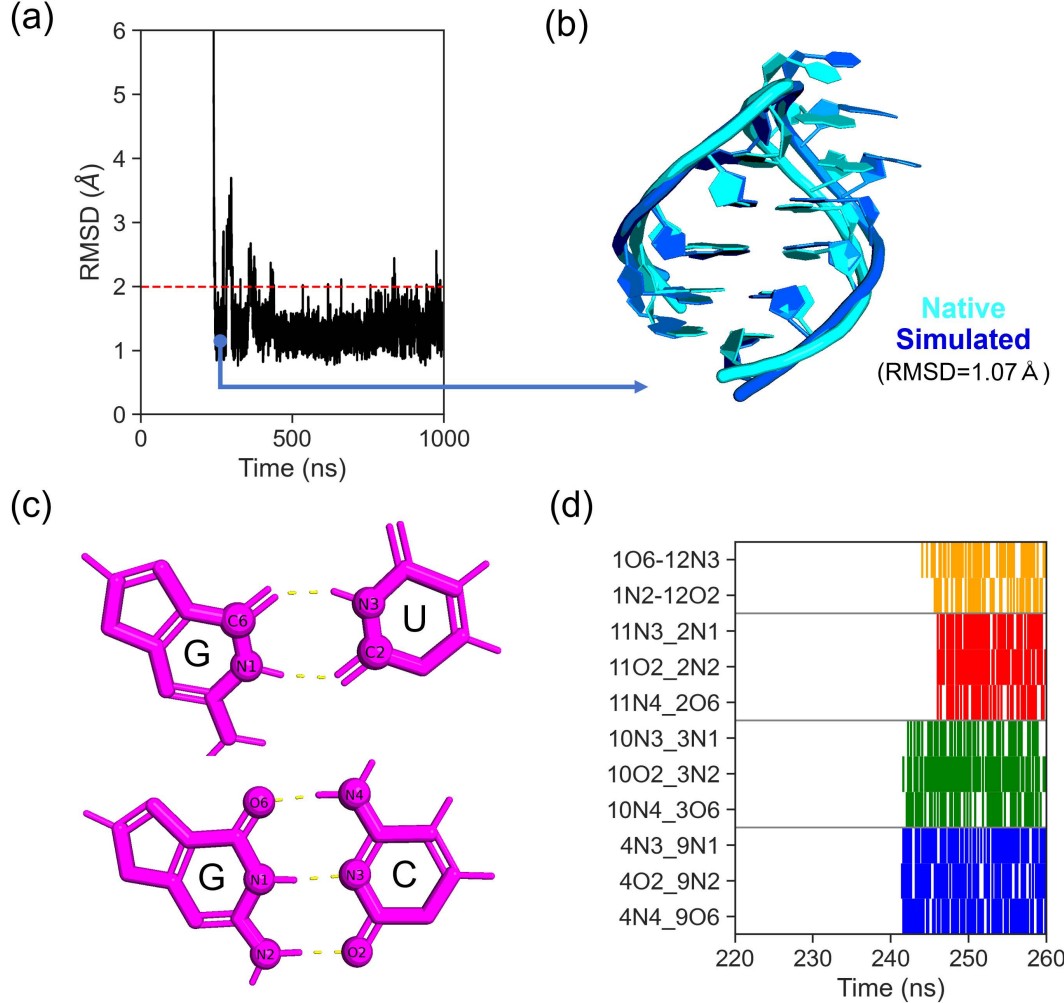

**Fig 2. Free folding simulations of RNA hairpin 1ZIH.** (a) Temporal evolution of RMSD in a successful folding trajectory (threshold: < 2 Å; red dashed line). 1-µs folding-phase segment is displayed to highlight native state attainment. (b) Structural superposition of the simulated folded conformation (blue) and native structure (cyan) with RMSD = 1.07 Å. (c) Hydrogen bond patterns (yellow dashed lines) in G-U and G-C base pairs. (d) Time-dependent formation of 11 hydrogen bonds in the trajectory from (a). A zoomed-in view reveals the sequential bond formation order.

these CTF simulations, we identified 3, 4, and 2 trajectories with succeed folding events, respectively (S5-10 Fig). Representative folding trajectories for each rate are illustrated in Figs 3a–c, with corresponding base-pairing or hydrogen-bond formation dynamics detailed in Figs 3d–f.

Comparative analysis of Figs 2 and 3 reveals identical final folded conformations between CTF (across all rates) and FF, yet divergent folding pathways characterized by altered base-pairing sequences. Given the limited number of observed pathways (14 in total), we classified trajectories into two distinct folding routes (Fig 4a):

PATH I: Initial formation of the 4C-9G pair adjacent to the loop region, followed by sequential establishment of other pairs.

PATH II: Priority formation of distal pairs (e.g., 2G-11C), with 4C-9G pairing occurring later.

Quantitative pathway distribution analysis (Fig 4b) shows that, in the FF simulations, 4 trajectories belong to PATH I and 1 to PATH II, indicating a preference for PATH I. In contrast, the CTF-1, CTF-10, and CTF-100 simulations exhibit 0/3, 1/3, and 1/1 trajectories assigned to PATH I/PATH II, respectively, collectively revealing a preference for PATH II. Notably, the folding pathway exhibited elongation-rate dependence, with faster transcription rates favoring PATH II emergence, suggesting kinetic modulation of folding pathway accessibility. However, FF—which can be conceptualized as CTF with

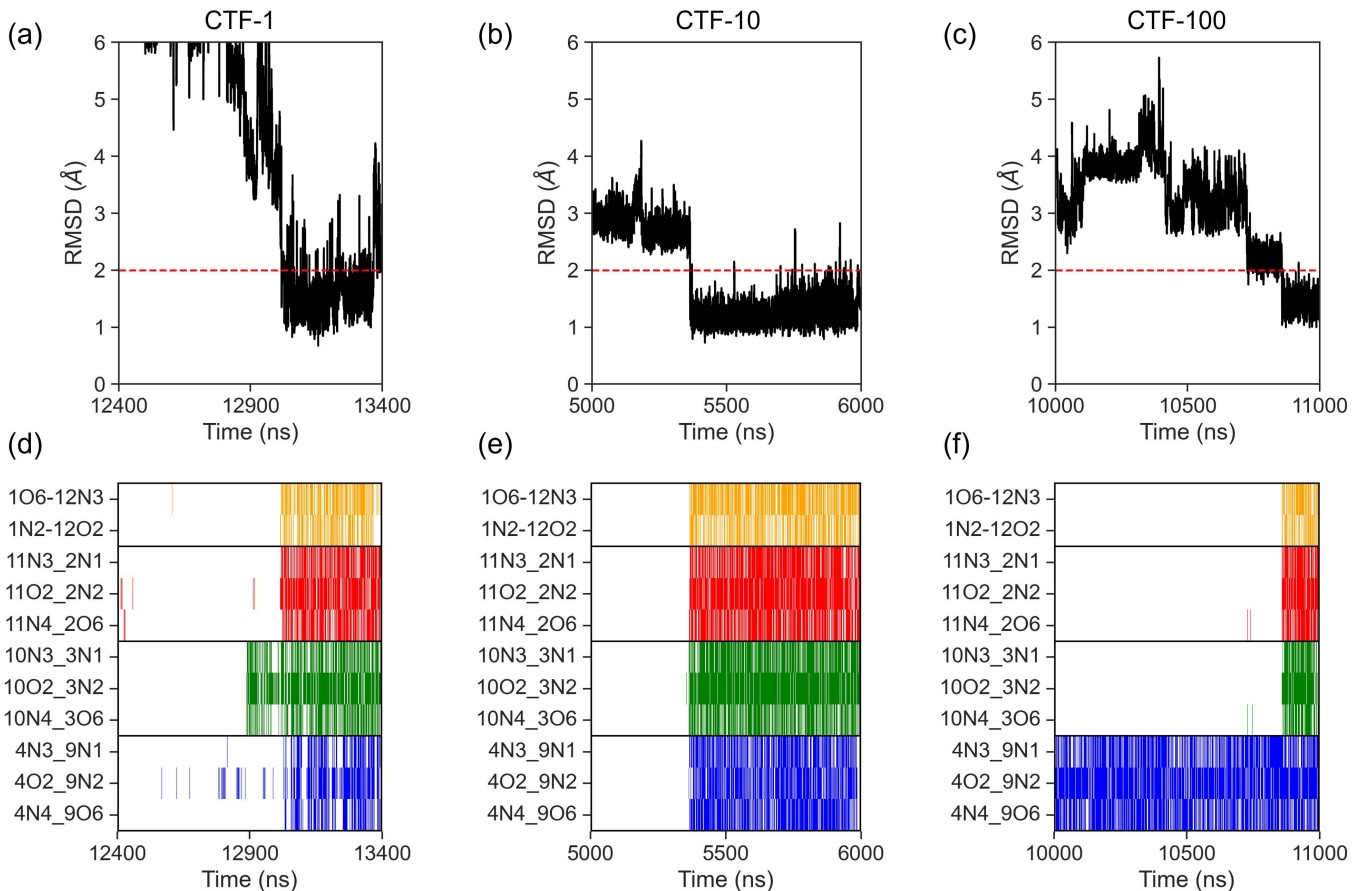

**Fig 3. Co-transcriptional folding simulations under varied transcription elongation rates.** (a-c) Temporal evolution of RMSD during co-transcriptional folding at 1/10/100 ns/nt (CTF-1/10/100). (d-f) Hydrogen bond formation chronology in (a-c). 1-µs folding-phase segments are displayed to highlight native state attainment.

(a)

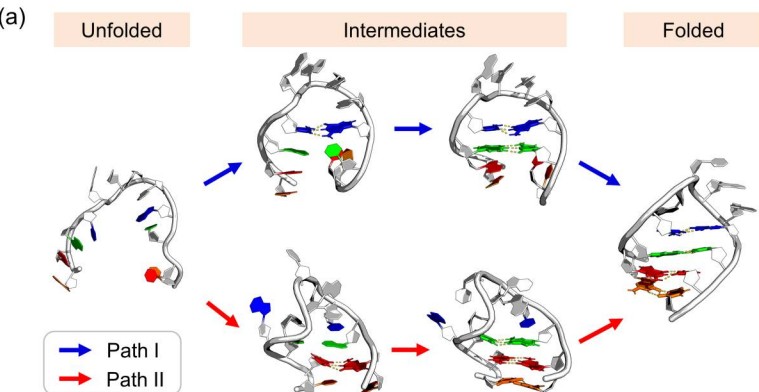

Unfolded Intermediates Folded

→ Path I
→ Path II

(b)

The number of trajectories that sampled PATH I or II in FF and CTF simulations

| Simulation (Total trajectories) | FF (60) | CTF-1 (12) | CTF-10 (12) | CTF-100 (12) |
|---|---|---|---|---|
| PATH I | 4 | 0 | 1 | 1 |
| PATH II | 1 | 3 | 3 | 1 |

**Fig 4. Folding pathways of 1ZIH.** (a) Two distinct folding pathways (PATH I and PATH II) observed in the simulations of 1ZIH, which primarily differ in the 4C-9G pair forms first or later. (b) The counts of these two folding pathways in free folding (FF) simulations and co-transcriptional folding (CTF) simulations at different transcription speeds.

an infinitely fast transcription rate—shows a preference for PATH I. There are two possible hypotheses to reconcile these observations:

1. Sampling limitations: Our current simulations captured an insufficient number of productive folding events for certain conditions. For example, CTF-100 (slowest rate) yielded only two successful folding trajectories, precluding robust statistical analysis of transcription rate effects across the full spectrum (Fig 4b).

2. Non-monotonic impact of transcription rate: The relationship between transcription rate and folding pathway preference may not be linear. We hypothesize an evolutionarily optimized critical transcription rate that maximizes folding efficiency. Deviations from this optimal rate may alter pathway preferences or reduce folding efficiency.

### 3.3 Co-transcriptional folding alters pathways via sampling of rare conformational states

While our comparative analysis revealed pathway divergence between CTF and FF for 1ZIH, as discussed in the previous subsection, the physical mechanism underlying this phenomenon remains unresolved. A plausible hypothesis is that CTF provides a non-equilibrium initial conformation upon transcription completion, establishing a distinct preference for folding pathways during the subsequent post-transcriptional folding compared to FF.

To test this hypothesis, we reconstructed the conformational landscape of 1ZIH using RMSD and radius of gyration (RoG) as orthogonal reaction coordinates. As shown in Fig 5, CTF conformations sampled across all three transcription speeds populate regions of smaller RMSD and RoG compared to FF. Since the intermediates of PATH II are inherently more compact than those of PATH I, these regions likely represent PATH II intermediates, thereby rationalizing the preferential folding of CTF along PATH II. The same conclusion also holds when all trajectories are included (S11 Fig). A representative CTF-100 trajectory revealed that transcription termination yields an intermediate conformation featuring two adjacent non-native base pairs (S12 Fig), which subsequently rearrange into native pairs during folding. This specific

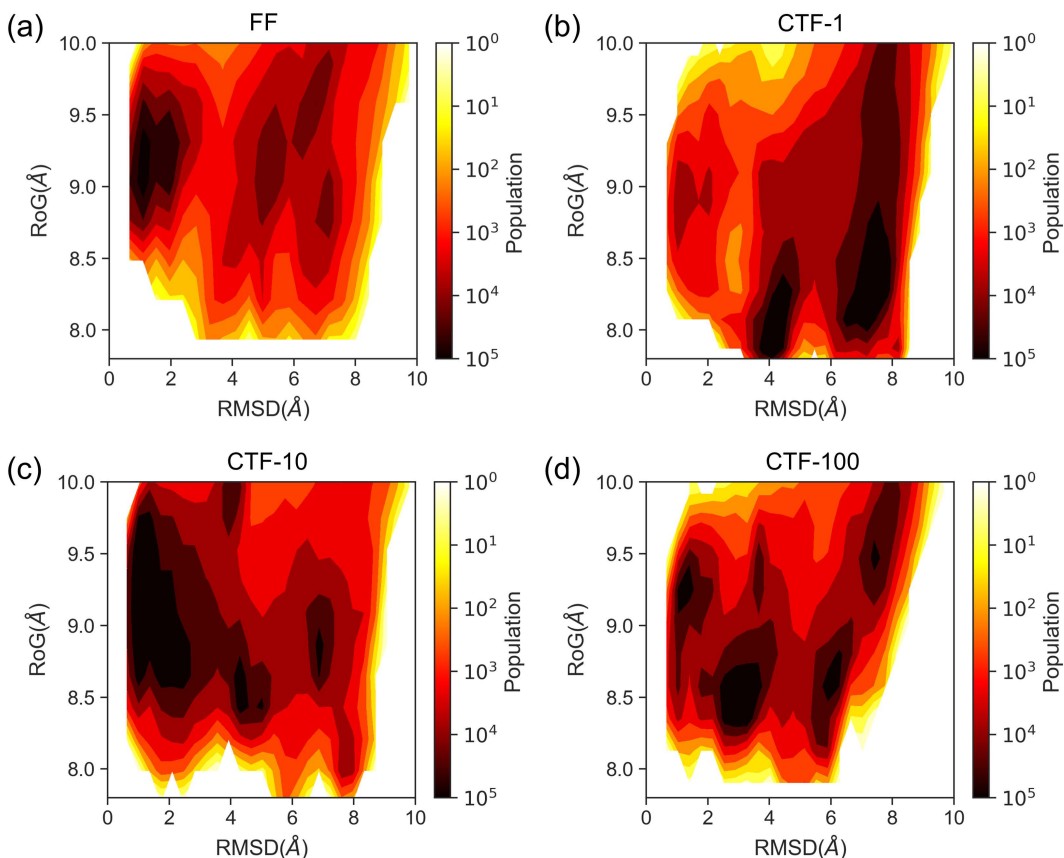

**Fig 5. Conformational distributions in 1ZIH simulations.** (a) Conformational distribution in free folding (FF) simulations. The reaction coordinates for the conformational space are the root-mean-square deviation (RMSD) of the stem region and the radius of gyration (RoG) of the entire 1ZIH molecule. (b-d) Conformational distribution in co-transcriptional folding (CTF) simulations at a transcription speed of 1, 10 and 100 ns/nt. The conformational distributions are built only on trajectory data from successful folding.

intermediate was not observed in any FF trajectory; nevertheless, non-native-to-native transitions of base pairs do occur in FF folding, as illustrated in S13 Fig.

Given that CTF can be conceptualized as FF initiating from a specialized initial structure, we systematically examined post-transcription conformations. Fig 6a displays the conformational distributions at $T_F = 0$ ns (the free folding time, defined as the time elapsed after transcription completion) for all trajectories. Crucially, slower transcription yields more pronounced differences between CTF and FF distributions at $T_F = 0$ ns. Tracking conformational evolution at later timepoints ($T_F = 10$, 100, and 1000 ns; Figs 6b-d), we observe that while differences between FF and CTF distributions diminish over time, the discrepancy still persists. Additionally, we found that at $T_F = 0$ ns, the slower the transcription speed, the greater the difference between the CTF and FF conformation distributions. At later times (i.e., $T_F > 10$ ns), the CTF-10 ensemble displays the smallest RMSD and RoG values. This heightened compactness correlates with CTF-10 achieving the highest folding success rate (4 out of 12 trajectories reaching the native state), suggesting that transcription speed exerts a non-monotonic influence on folding efficiency.

Collectively, these results elucidate a pathway regulation mechanism for CTF: Transcription generates compact initial conformations rarely sampled in FF. Consequently, CTF establishes a distinct preference for folding pathways during the subsequent post-transcriptional folding compared to FF, and this pathway preference is modulated by transcription speed.

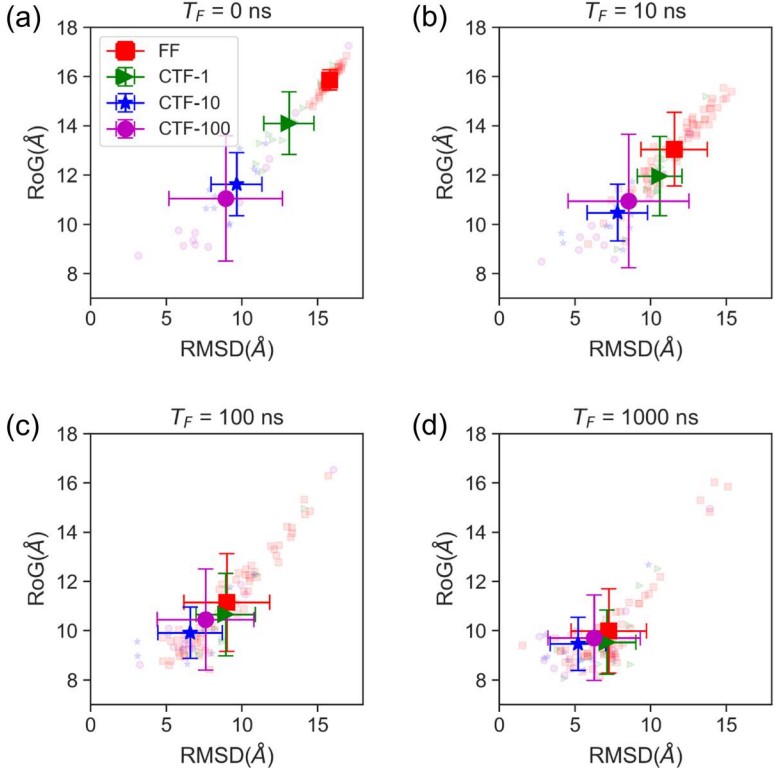

**Fig 6. Conformational evolution of 1ZIH.** (a–d) Conformational distributions of 1ZIH under four simulation scenarios (FF, CTF-1, CTF-10, and CTF-100) at different free-folding times ($T_F = 0$, 10, 100, and 1000 ns). In FF simulations $T_F$ equals the actual simulation time, whereas in CTF simulations $T_F$ is obtained by subtracting the transcription duration ($11\Delta t$) from the actual simulation time. The reaction coordinates describing the conformational space are the root-mean-square deviation (RMSD) of the stem region and the radius of gyration (RoG) of the entire 1ZIH molecule. Small semi-transparent symbols denote the conformations of 1ZIH at $T_F$; red squares, green triangles, blue stars, and purple circles correspond to the FF, CTF-1, CTF-10, and CTF-100 simulations, respectively. Large opaque symbols represent the means of each group, with error bars indicating the standard deviations.

## 4 Discussions

In this study, we developed a simplified scheme for simulating the RNA CTF process. Utilizing this scheme, we investigated the CTF process of the 1ZIH hairpin and compared it with FF. Our findings indicate that although CTF does not alter the final product or native state of 1ZIH, it may modify the preference of folding pathway.

Specifically, the FF simulations indicate a preference for PATH I, characterized by the 4C-9G pair near the loop forms first, followed by the formation of other pairings. This zipper-like folding mechanism is also observed in proteins [35]. In contrast, the CTF-1, CTF-10, and CTF-100 simulations exhibit a preference for PATH II, denoted by the pair distant from the loop form first, and the 4C-9G pair forms later. The observation that CTF can alter the preference of folding pathways aligns with the results of previous computational studies [15]. Moreover, the regulatory effect of transcription speed on the proportion of folding pathways has also been reported in studies of co-translational folding in proteins [36–39]. For example, Till et al. [40] demonstrated that the ribosome and trigger factor collaboratively compact nascent proteins to accelerate folding. Despite differences between biological systems (protein vs. RNA), these analogous findings strongly support our proposed pathway regulation mechanism in CTF. Finally, we propose a possible explanation for the mechanism by which CTF modifies folding pathways: viewing CTF as a form of FF that possess more compact initial conformations, which are difficult to sample in convention FF and direct RNA towards different folding pathways. Theoretically, the present study challenges the inherent assumption of "native-state dominance" in the traditional RNA folding energy landscape

theory [41]. We have found that, even for a simple hairpin structure, CTF can overcome free energy barriers through a nonequilibrium synthesis process to access pathways that are negligible in probability within FF. This implies that the conformational diversity of native RNA is determined not only by thermodynamic stability but is also subject to the encoded regulation of transcriptional kinetic parameters. However, GNRA tetraloops *in vivo* typically fold within larger structural contexts (e.g., riboswitches or ribozymes), where flanking sequences may influence folding kinetics. We will employ more complete transcripts in our future work.

While pioneering coarse-grained studies [15,18] revealed different folding pathway between CTF and FF, our all-atom MD framework provides unprecedented resolution in capturing non-native intermediate states inherent to RNA CTF. Monte Carlo approach [15], constrained by predefined base-pairing rules, lacked atomic-scale insights into transient fluctuations (e.g., backbone torsional strain or metastable hydrogen bond) that kinetically gate pathway selection. Although the hybrid method [18] integrated NMR constraints, its simplified energy function insufficiently modeled non-canonical base-stacking interactions critical for early folding intermediates. In contrast, our simulations identified CTF-specific high-energy intermediates that often averaged out in coarse-grained approximations.

However, the present study has several methodological limitations. First, although the simplified CTF simulation scheme we proposed can be extended to other RNAs, it only considers the characteristic of sequential synthesis and folding. The interactions between the nascent RNA chain and RNA polymerase are not taken into account [25]. Second, in real cells, the transcription speed of RNA is typically tens of nucleotides per second, whereas in our simulations, the transcription speed is 1–100 ns per nucleotide [3], which is much faster than the actual transcription speed. Moreover, the real transcription speed of RNA is dynamically variable and sometimes includes transcriptional pausing [16]. Due to the limitations of the simulation methods, these factors have also been overlooked. Next, despite accumulating over 800 microseconds of all-atom simulation trajectories, we only observed 14 complete folding pathways in total, with only 2–5 pathways for each type of simulation. Therefore, the reliability of our conclusions at the statistical level is insufficient. Finally, we only investigated the simplest RNA hairpin, 1ZIH, which is only a part of a complete RNA molecule. In addition, the simulations performed in this study applied only one combination of force field, ion, and solvent models, and different combinations may significantly alter the folding kinetics of biomolecules [42–48]. Therefore, whether the results and conclusions regarding CTF in this study are universal requires more extensive and systematic research. These improvements will be included in our future works.

In summary, by developing an all-atom MD framework to simulate RNA CTF, this study reveals transcription rate-dependent pathway diversification that reshapes energy landscape exploration and challenges classical thermodynamic paradigms, offering kinetic insights into RNA functional regulation.

## Supporting information

**S1 Fig. Comparison of the distribution of RMSD and RoG using different water buffer sizes.**
(TIF)

**S2 Fig. An example of a completely folded trajectory.**
(TIF)

**S3 Fig. Temporal evolution of RMSD in 60 FF simulations.**
(TIF)

**S4 Fig. Temporal evolution of 11 hydrogen bonds in 60 FF simulations.**
(TIF)

**S5 Fig. Temporal evolution of RMSD in 12 CTF-1 simulations.**
(TIF)

**S6 Fig. Temporal evolution of 11 hydrogen bonds in 12 CTF-1 simulations.**
(TIF)

**S7 Fig. Temporal evolution of RMSD in 12 CTF-10 simulations.**
(TIF)

**S8 Fig. Temporal evolution of 11 hydrogen bonds in 12 CTF-10 simulations.**
(TIF)

**S9 Fig. Temporal evolution of RMSD in 12 CTF-100 simulations.**
(TIF)

**S10 Fig. Temporal evolution of 11 hydrogen bonds in 12 CTF-100 simulations.**
(TIF)

**S11 Fig. Conformational distributions in 1ZIH simulations.**
(TIF)

**S12 Fig. The structural ensembles of CTF simulations when transcription terminates.**
(TIF)

**S13 Fig. An example of the trajectory of non-native-to-native base pair transitions in FF simulations.**
(TIF)

## Author contributions

**Conceptualization:** Peng Tao.

**Formal analysis:** Peng Tao, Yunda Si, Jie Xia.

**Funding acquisition:** Peng Tao, Yi Xiao.

**Investigation:** Peng Tao, Yunda Si, Jie Xia.

**Methodology:** Peng Tao, Yunda Si, Jie Xia.

**Project administration:** Peng Tao, Yi Xiao.

**Resources:** Peng Tao.

**Software:** Peng Tao.

**Supervision:** Peng Tao, Yi Xiao.

**Validation:** Yunda Si, Jie Xia.

**Visualization:** Peng Tao, Yunda Si, Jie Xia.

**Writing – original draft:** Peng Tao, Yunda Si, Jie Xia.

**Writing – review & editing:** Peng Tao, Yi Xiao.

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
