## [Decision Letter · Decision Letter 0]

12 May 2025

Transcription Reshapes RNA Hairpin Folding Pathways Revealed by All-Atom Molecular Dynamics Simulations

PLOS Computational Biology

Dear Dr. Tao,

Thank you for submitting your manuscript to PLOS Computational Biology. After careful consideration, we feel that it has merit but does not fully meet PLOS Computational Biology's publication criteria as it currently stands. Therefore, we invite you to submit a revised version of the manuscript that addresses the points raised during the review process.

Please submit your revised manuscript within 60 days Jul 12 2025 11:59PM. If you will need more time than this to complete your revisions, please reply to this message or contact the journal office at ploscompbiol@plos.org. Please include the following items when submitting your revised manuscript:

We look forward to receiving your revised manuscript.

Kind regards,

Shi-Jie Chen

Academic Editor

PLOS Computational Biology

Nir Ben-Tal

Section Editor

PLOS Computational Biology

**Journal Requirements:**

At this stage, the following Authors/Authors require contributions: Peng Tao, Yunda Si, Jie Xia, and Yi Xiao. Please ensure that the full contributions of each author are acknowledged in the "Add/Edit/Remove Authors" section of our submission form.

5) In the online submission form, you indicated that "The origin trajectory data are available from the corresponding author upon request."  All PLOS journals now require all data underlying the findings described in their manuscript to be freely available to other researchers, either

1. In a public repository

2. Within the manuscript itself

3. Uploaded as supplementary information.

**Reviewers' comments:**

Reviewer's Responses to Questions

Reviewer #1: RNA folding during transcription is a complex process, collectively influenced by many factors, such as the speed of elongation, site-specific pausing and interactions between the nascent RNA chain and the RNA polymerase. This manuscript by Tao et. al. introduces a simplified all-atom molecular dynamic framework for RNA cotranscriptional folding simulations. Although it focuses on the exploration of transcription speed in RNA hairpin cotranscriptional folding, the method it provides for atomic-scale mechanistic studies provides a new approach for understanding the intricate dynamics of RNA folding. This study is intriguing, however, there are still several inquiries that need to be addressed.

(1) RNA free folding (FF) begins with a completely unstructured coil, while RNA cotranscriptional folding (CTF) involves a chain growth process where the RNA folds as it elongates. By the time the full chain is synthesized, its structure is not fixed, but it typically differs from the coil structure. The subsequent folding process represents a refolding of the entire chain. Thus, the distinction between FF and CTF can be understood as a difference in initial structure. In the article, the authors chose different simulation times for FF and CTF (3-4 μs vs. 15 μs). It is advisable to sample FF for the same duration as CTF, while also noting that this 15 μs should not include the chain elongation time. This adjustment would ensure a more reasonable comparison between the two folding processes.

(2) FF has 60 independent simulations, while each transcription speed for CTF only provides 10. It is advisable to increase the number of CTF simulations. This would enhance the statistical significance of the different folding pathways, allowing for a more robust comparison and better understanding.

(3) Figure 5 is very important, but it needs to include the cases for CTF-1 and CTF-10. Additionally, the discussion in Section 3.3 regarding the impact of transcription speed on CTF folding is not sufficiently thorough. It would also be beneficial to add annotations for the initial structures (coil for FF and post-transcription structures for CTF) in Fig. 5. This would enhance the ability to observe how transcription speed influences the initial conformations and folding processes.

Additional minors:

(4) On p6 “Systems were solvated in TIP3P (12 Å buffer)”: A buffer of this size may be limited, as it might not adequately accommodate the structural changes and dynamics during the folding process.

(5) On p10 “The criteria for hydrogen bond formation were based on a maximum donor-acceptor distance of 3.5 Å and a minimum donor-proton-acceptor angle of 120°.” It would be beneficial to include references that support these criteria.

(6) On p11, the reference to “see Supplementary Fig. S2 for the others” should be corrected to Fig. S5.

(7) On p13, “with slower transcription rates favoring PATH II emergence” seems inconsistent with the data presented in Fig. 4b, where CTF-1, which has the fastest transcription rate, favors PATH II. Additionally, FF can be considered a special case of cotranscriptional folding with an infinitely fast rate, yet it favors PATH I. Authors need to clarify these observations.

(8) The term "inaccessible" is too absolute. It would be more accurate to state that the structures and folding pathways sampled by CTF are also accessible to FF; however, the probabilities of sampling these structures and pathways differ.

Reviewer #2: While this manuscript was interesting, engaging to read, deliberate in its approach to investigate FF vs. CTF, I find that it is preliminary and makes several assumptions that are either testable by further analysis of the current dataset, or testable by additional simulations. I really hope the authors refine this work, as I would be really interested to read a much deeper analysis of their simulation set, as would the rest of the RNA dynamics community.

The authors introduce a bias in the form of a rigid plane to mimic the behavior of nucleotides directly after transcription. No interactions are allowed between the nucleotides on either side of the plane. This is an elegant solution to the problem of modeling this with “accessory” proteins, but I’m not sure it’s strictly correct – GNRA TLs aren’t necessarily formed in a single 12mer transcript, but as part of larger structures. The authors should discuss the implications of this.

Using only the stem as a measure of “folded” is incorrect here – the 2 Angstrom RMSD cutoff for “folded” is a particularly low bar considering one wouldn’t need to actually fold the loop correctly at all to achieve it. I am concerned that the graphical abstract is misleading – do the 2 or 3 simulations which “fold” (i.e., attain the 11 h-bonds in the stem) ever display the correct loop conformation? If not, having an arrow pointing to the “native” fold is incorrect. If they do attain the correct loop fold, you should explain that as part of the manuscript.

The sampling for each situation/case was investigated extremely well.

I appreciate the detail in h-bonding criteria in the Methods section.

Figure 1 is very nice and I like that all the colors are consistent there and throughout the text.

As written, Figure 4B Table title is confusing. This table could be clearer by adding the number out of the total number of trajectories sampled (3/60). If you have only sampled each path once, then the results are different than if you mean that Path I is the only path that is sampled, and it is sampled in each trajectory.

Page 13, line 12: “These results establish that transcription-coupled folding enables pathways inaccessible during equilibrium refolding.” You have not proven this with these simulations, so this assertion is not “established.” You can change the language here to “These results suggest more work should be done to see if CTF pathways are accessible during FF.” You have observed that the FF pathways do not sample the pathways observed in the CTF simulations – but you have not proven that it is not *possible*. You could seed the FF simulations with an intermediate to test the CTF pathway, you could find a reaction coordinate and make an argument energetically, you could search the other trajectories. More work needs to be done to “establish” it, especially with the limited number of paths that the current sampling returned.

Figure 5: It is unclear what data you are incorporating into your conformational distribution plots. I think it would be great to have one set of plots where you include ALL simulated data, and one set of plots where you exclude all trajectories that did not fold. Also, please include an explanation for the arrows in the caption of this Figure.

Page 15, line 17-19: “In CTF simulations, in addition to the pathway present in FF simulations, we observed a folding pathway that do not exist in FF, where the pair distant from the loop form first, and the 4C-9G pair forms last.” This statement is only partially true – the h-bond vs. time plots in the SI Fig. 5 show that in multiple FF simulations, h-bonds form in the base pairs further away from the loop (yellow, red and green). The full TL never “folds,” but I’m not sure you’re adequately addressing all of the sampled data here. It might be interesting to understand what happens in the FF simulations when far h-bonds form, but the loop ultimately does not “fold.” It could certainly explain differences between each (FF and CTF) pathway.

Page 16, line 3-5: “Finally, we propose a possible explanation for the mechanism by which CTF modifies folding pathways: viewing CTF as a form of FF that may possess rare initial conformations, which are difficult to sample in FF and direct RNA towards different folding pathways.” This is testable from the simulations you have run – why not look at the distribution of “starting” structures and compare using a combined clustering analysis or 2d RMS? You can also test whether any of the FF simulations sample any of the CTF structures by comparing all structures with a 2D RMS plot. You could also measure backbone dihedrals to ascertain whether structures in the CTF are not seen in the FF simulations.

Reviewer #3: In this manuscript, the authors applied all-atom MD simulation to study the effect of transcription on the folding pathways of an RNA hairpin. They implement a movable rigid plane to prevent non-physical nteractions between synthesized and unsynthesized nucleotides and obtained that RNA transcription may diversify folding dynamics. However, the manuscript needs minor revision to improve the readability.

1) Page 7,the initial structure should be presented, is Fig1b the intial structure?

2) Page 6, is the periodic boundary conditions used? If it is, as the initial structure is extended in x-direction, what is thickness of solvent layer at each direction?

3) Fig5, is the distribution of one pathway or all simulations? If it is of one pathway, as there are 3 pathways for FF folding, is the distribution same for each pathway? and there are two kinds of pathway for CFT folding, Fig5b represents which pathway?

4) FigS9, are the hydrogen bonds of the non-native base pair in the structure of the intermediate state formed? Could the structure be found in the FF simulations except the 3 successful folding simulations?

**Have the authors made all data and (if applicable) computational code underlying the findings in their manuscript fully available?**

Reviewer #1: None

Reviewer #2: **No: ** The significant structures could be deposited in PDB format.

Reviewer #3: None

PLOS authors have the option to publish the peer review history of their article (what does this mean? ). If published, this will include your full peer review and any attached files.

**Do you want your identity to be public for this peer review?** For information about this choice, including consent withdrawal, please see our Privacy Policy .

Reviewer #1: No

Reviewer #2: No

Reviewer #3: No

**Figure resubmission:**

**Reproducibility:**



---

## [Decision Letter · Decision Letter 1]

28 Aug 2025

Dear Dr. Tao,

We are pleased to inform you that your manuscript 'Transcription Reshapes RNA Hairpin Folding Pathways Revealed by All-Atom Molecular Dynamics Simulations' has been provisionally accepted for publication in PLOS Computational Biology.

Best regards,

Shi-Jie Chen

Academic Editor

PLOS Computational Biology

Nir Ben-Tal

Section Editor

PLOS Computational Biology

Reviewer's Responses to Questions

**Comments to the Authors:**

Reviewer #1: Thank you for providing the revised manuscript. I have reviewed the changes and have no further comments.

Reviewer #2: The authors addressed my comments and I believe this manuscript is suitable for publication.

Reviewer #3: All of my concerns have been clarified.

**Have the authors made all data and (if applicable) computational code underlying the findings in their manuscript fully available?**

Reviewer #1: None

Reviewer #2: Yes

Reviewer #3: None

PLOS authors have the option to publish the peer review history of their article (what does this mean? ). If published, this will include your full peer review and any attached files.

**Do you want your identity to be public for this peer review?** For information about this choice, including consent withdrawal, please see our Privacy Policy .

Reviewer #1: No

Reviewer #2: **Yes: ** Christina Bergonzo

Reviewer #3: No

---

## [Editor Report · Acceptance letter]

PCOMPBIOL-D-25-00786R1

Transcription Reshapes RNA Hairpin Folding Pathways Revealed by All-Atom Molecular Dynamics Simulations

Dear Dr Tao,

I am pleased to inform you that your manuscript has been formally accepted for publication in PLOS Computational Biology. Your manuscript is now with our production department and you will be notified of the publication date in due course.

With kind regards,

Judit Kozma
